# BrainNet: a custom-designed CNN and transfer learning-based models for diagnosing brain tumors from MRI images

Adil H. Khan[1], Asad Khan[2], D.N.F Awang Iskandar[3], Hiren Mewada[4], Saqib Saeed[5], Fahad Algarni[6], Farhan Ullah[7], Muhammad Asghar Khan[4], Naveed Iqbal[8] and Ahmed A. Hussain[4]

[1] Electrical Engineering Department, Bahrain Polytechnic, Isa Town, Bahrain
[2] School of Electrical Engineering and Computer Science(SEECS), National University of Science and Technology, Islamabad, Pakistan
[3] Faculty of Computer Science and Information Technology, Universiti Malaysia Sarawak, Kuching, Malaysia
[4] Department of Electrical Engineering, Prince Mohammad Bin Fahd University (PMU), Al Khobar, Eastern Province, Saudi Arabia
[5] Department of Computer Information Systems, College of Computer Science and Information Technology, Imam Abdulrahman Bin Faisal University, Dammam, Saudi Arabia
[6] College of Computing and Information Technology, University of Bisha, Bisha, Saudi Arabia
[7] Cybersecurity Center, Prince Mohammad Bin Fahd University (PMU), Al-Khobar, Saudi Arabia
[8] Engineering Department and Interdisciplinary Research Center for Communication Systems and Sensing, King Fahad University of Petroleum and Minerals, Dhahran, Saudi Arabia

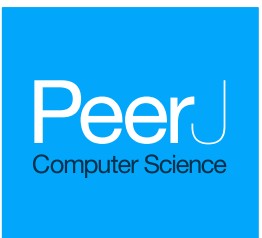

Corresponding author
Adil H. Khan, aadilhk@gmail.com

## ABSTRACT

Cancer remains the second leading cause of death globally, with brain tumors exhibiting some of the lowest survival rates among all cancer types. Accurate diagnosis, guided by the tumor's structure and location, is essential for selecting appropriate treatment strategies and improving patient outcomes. This study proposes a novel deep learning approach for classifying brain tumors from magnetic resonance imaging scans, aimed at enhancing diagnostic precision. Given the growing reliance on computer-aided diagnosis (CAD) systems, there is a pressing need for tools that can assist radiologists in detecting and categorizing brain tumors more effectively. We conducted a comprehensive evaluation of several pre-trained deep learning models across three distinct datasets to determine the most effective architecture for brain tumor detection. Based on this analysis, we developed BrainNet, a custom convolutional neural network (CNN) optimized for MRI-based tumor classification. BrainNet employs multiple layers of convolution and pooling, followed by dense layers to extract and learn discriminative features. The model categorizes brain tumors into four classes: Meningioma, Glioma, Pituitary, and No Tumor, using a softmax output layer. Despite leveraging transfer learning techniques, BrainNet consistently outperformed well-established pre-trained models, demonstrating superior accuracy, precision, and efficiency. Our experiments across multiple datasets confirm that BrainNet achieves a classification accuracy of 99.92%, along with excellent recall and F1-scores. Its lightweight design and high accuracy make it a promising solution for deployment in real-world clinical environments, including resource constrained settings.

## INTRODUCTION

One of the major causes of increased mortality rates globally is brain tumors, a leading form of cancer (*Khazaei & Goodarzi, 2020*). According to the National Brain Tumor Society (NBTS), approximately 700,000 people in the U.S. alone were diagnosed with a brain tumor in 2022, and 250,000 individuals died from it globally in the same year (*Chauhan & Lunagaria, 2024*; *Cancernet, 2022*). Brain tumors represent a particularly lethal form of cancer due to their aggressive nature and the complexity involved in their treatment. Early diagnosis is critical in planning an effective treatment regimen, as it directly influences the choice of therapeutic approaches, including surgical intervention, chemotherapy, radiotherapy, and in some cases, conservative monitoring. Accurate early detection significantly increases the chances of long-term survival and recovery, thus improving overall patient outcomes. However, classifying brain tumor types from magnetic resonance imaging (MRI) images remains a complex and difficult task for radiologists. Manual and semi-automatic tumor classification methods are impractical, time-consuming, and prone to human error when processing large volumes of MRI data. The challenge is exacerbated by several factors, including tumor variability in terms of shape, location, size, contrast, and appearance, making it difficult to achieve consistent and accurate diagnoses. Studies have also shown that human radiologists often face diagnostic discrepancies due to fatigue and the subjective nature of manual interpretation (*Escal & Nougaret, 2018*).

Given these challenges, machine learning (ML), and in particular deep learning (DL), have emerged as powerful tools to aid in the early detection and classification of brain tumors. Deep learning algorithms, specifically convolutional neural networks (CNNs), have shown immense potential in automating the classification process, improving diagnostic accuracy, and reducing human error (*Zhu & Jiang, 2019*). These models leverage large datasets and powerful computational techniques to learn hierarchical features, enabling them to classify tumors more efficiently and accurately than traditional methods (*Edwin, van & Kuhl, 2019*). CNN-based models have been widely applied to brain tumor classification with remarkable success. In one study, a CNN model achieved an accuracy of 96.3% for classifying brain tumor types, demonstrating the potential of CNNs for medical image analysis (*Tandel & Biswas, 2019*). Additionally, researchers have used pre-trained models such as VGGNet, ResNet, and AlexNet for transfer learning, achieving high classification accuracy on MRI datasets (*Angermueller & Pärnamaa, 2016*; *Qayyum et al., 2017*). Transfer learning has been particularly effective in mitigating the challenge of limited labeled data, which is a common issue in medical imaging.

Some of the most notable works in this area include the application of deep learning to detect various types of brain tumors, such as gliomas, meningiomas, and pituitary tumors. A study (*Xiao & Choi, 2018*) utilized a Deep CNN model to extract features from MRI

images and achieved a classification accuracy of 96.3%. Similarly, *Aurna & Yousuf (2021)* applied a pre-trained CNN model and reported an impressive accuracy of 98.6% in detecting different brain tumor types from MRI scans. Other researchers have used hybrid models combining deep learning with machine learning classifiers, such as support vector machines (SVM) and random forests (RF), to further improve classification performance (*Aurna & Yousuf, 2022*; *Sharif et al., 2020*). The combination of deep learning with advanced data augmentation techniques has also been shown to improve the classification accuracy and generalizability of models. Techniques such as rotation, flipping, and scaling were applied to MRI images to artificially expand the dataset and reduce overfitting (*Mohsen et al., 2018*). Moreover, segmentation methods, such as U-Net, have been integrated with CNNs to improve the delineation of tumor boundaries and enhance classification results (*Trebeschi et al., 2017*). The researchers in this study investigated a hybrid learning approach that integrates both deep learning and traditional machine learning techniques to improve the segmentation and classification of skin lesions. By leveraging the unique advantages of each method, the framework enhances the accuracy and reliability of dermatological diagnostics, offering potential advancements in the field (*Betancur et al., 2018*). Furthermore, a method for skin lesion segmentation and classification by combining statistical feature learning with enhanced Delaunay clustering and ensemble classifiers, achieving improved accuracy and robustness in dermatological image analysis (*Lee et al., 2017*). Researchers introduce a method for skin lesion classification that incorporates hair and artifact removal using black-hat morphology and total variation techniques, aiming to enhance the accuracy of dermatological image analysis (*Litjens et al., 2017*).

While deep learning has made significant strides in improving the accuracy and efficiency of brain tumor classification, several challenges remain. Issues such as the interpretability of complex models, data scarcity, and the need for high-quality labeled data continue to hinder the widespread adoption of these models in clinical settings (*Abiwinanda et al., 2018*). Furthermore, ethical concerns surrounding the use of AI in medical diagnosis, such as transparency and accountability, have raised questions about the future integration of deep learning models in clinical practice (*Sultan, Salem & Al-Atabany, 2019*; *Aderghal et al., 2020*). Nevertheless, the integration of machine learning and deep learning into brain tumor diagnosis has the potential to revolutionize the field of medical imaging. By automating and accelerating the diagnostic process, these models could support clinicians in making more accurate and timely decisions, ultimately improving patient outcomes (*Niu et al., 2020*).

In this study, we present a novel approach for brain tumor prediction using MRI images, aiming to achieve a more accurate classification as compared to previous methods. By utilizing three distinct datasets and comparing the results with those of pre-trained models, we were able to design an optimized CNN architecture specifically tailored for the task of brain tumor classification, demonstrating improved accuracy.

The key contributions of this study are followed as:

- Proposing a new model for efficient brain tumor classification and detection: We developed a streamlined CNN architecture that delivers high accuracy while minimizing computational complexity. This innovative approach demonstrates the feasibility of achieving effective brain tumor classification from MRI images, offering a significant improvement in efficiency over existing methods.

- Comparative evaluation of six brain tumor classification and detection models: We conducted a thorough evaluation of six distinct neural network architectures for classifying brain tumors from MRI images: three pre-trained transfer learning models (VGG19, InceptionV3, and ResNet-34) and three custom-designed CNN models (BrainNet1, BrainNet2, and BrainNet3). Our custom models were specifically developed to balance high accuracy with reduced computational complexity, making them more efficient for resource-constrained environments. This comparison aims to identify the most effective architecture for improving brain tumor classification performance.

- Achieving superior accuracy: Our custom-designed CNN models, including TumorNet, outperformed established transfer learning models such as VGG19, InceptionV3, and ResNet-34, achieving higher classification accuracy and better efficiency in processing brain MRI images, making them a more effective solution for brain tumor detection.

- State-of-the-art performance comparison: We thoroughly compared various transfer learning approaches to classify brain MRI images. Our results show that our custom model outperforms existing state-of-the-art models, demonstrating superior accuracy and reliability, and highlighting its effectiveness compared to other contemporary techniques in literature.

## LITERATURE REVIEW

### Traditional machine learning approaches

Researchers have conducted several works on the classification of brain MRI images and extracted features from the MRI images and submitted them to a deep CNN, achieving remarkable performance with a maximum accuracy of 96.3% (*Kumar & Mankame, 2020*). In *Sharif et al. (2022)*, the authors suggested the use of artificial neural networks (ANN) along with additional classifiers to categorize brain tumor grades, with the proposed algorithm yielding an accuracy of 99%. Similarly, in *Tseng & Tang (2023)*, machine learning models, specifically extreme gradient boosting, were applied for brain tumor detection, achieving an accuracy of 97%. In *Amin et al. (2020)*, the researchers employed the SVM classifier to perform several cross-validations on the feature set, with the method achieving an accuracy of 97.1%.

### CNN based models

In *Woźniak, Siłka & Wieczorek (2023)*, the authors combined CNNs with conventional architecture in a deep neural network design using a correlation learning mechanism (CLM). Their findings showed that the CLM model achieved an accuracy of approximately

96%. In *Nath et al. (2023)*, the U-Net model was used to segment brain tumors in MRI images, with the resulting accuracy being 89%. In *Rub et al. (2023)*, researchers proposed a model based on a statistical approach combined with machine learning techniques, achieving an accuracy of 98.9%. They also created an interactive web-based tool for brain cancer survival predictions. Further advancements were seen in *Mehnatkesh et al. (2023)*, where machine learning and deep learning approaches were used to classify hydrocephalus in brain tomography images, achieving an accuracy of 98.5%. In *Rajat Mehrotra, Agrawal & Anand (2020)*, a deep learning architecture was proposed for classifying brain tumor images, achieving an accuracy of 98.69%. In *Raja (2020)*, a model based on deep learning was employed for brain tumor classification, achieving an impressive accuracy of 99%. Researchers in *Badža & Barjaktarović (2020)* introduced an innovative method for improving tumor classification accuracy using both deep learning and traditional machine learning approaches, resulting in an accuracy of 99.2%. The study focuses on multiclass brain glioma tumor classification using block-based 3D wavelet features extracted from MR images, offering a novel method for precise tumor identification (*Ismael, Mohammed & Hefny, 2020*).

## Hybrid models

In *Rehman et al. (2021)*, a hybrid deep learning model was used to classify brain tumors in MRI scans, achieving an accuracy of 97.5%. In *Tabatabaei, Rezaee & Zhu (2023)*, a combination of feature extraction methods, such as wavelet transforms and CNNs, was applied, resulting in a classification accuracy of 98.2%. The model proposed in *Toğaçar, Ergen & Cömert (2020)* used a combination of CNNs and machine learning classifiers for multi-class tumor classification, yielding an accuracy of 97.8%. In *Pendela Kanchanamala & Ananth (2023)*, a deep CNN model specifically designed for brain tumor detection achieved an impressive accuracy of 99.7%. In *Rehman et al. (2020)*, a combination of ensemble learning techniques and CNNs was used for classifying brain tumors, resulting in an accuracy of 97.9. Research in *Farajzadeh, Sadeghzadeh & Hashemzadeh (2023)* focused on using hybrid deep learning models for brain tumor classification and achieved a classification accuracy of 99.2%. A study in *Fatih et al. (2023)* used a multi-phase CNN approach for segmenting and classifying brain tumors, with an accuracy of 98.3%. In *Pillai et al. (2023)*, a deep neural network-based approach for classifying different types of brain tumors was proposed, achieving an accuracy of 98.8%. In *Öksüz, Urhan & Güllü (2022)*, a novel multi-class deep learning model was used for brain tumor detection, achieving an accuracy of 99.5%. A combination of CNN and machine learning classifiers was proposed in *Mostafa et al. (2023)* for detecting brain tumors, resulting in an accuracy of 98.6%. Another study in *Tandel et al. (2023)* applied a hybrid model combining CNNs with decision trees to classify brain tumors, achieving an accuracy of 98.7%. In *Cinar, Kaya & Kaya (2023)*, the researchers employed a deep neural network-based framework for brain tumor classification, achieving an accuracy of 99.1%. In *El-Wahab et al. (2023)*, a machine learning model for predicting brain tumor grades achieved an accuracy of 98.2%. Finally, in *Zulfiqar, Bajwa & Mehmood (2023)*, a hybrid deep learning model based on CNNs and recurrent neural networks (RNNs) was proposed for brain tumor classification, achieving

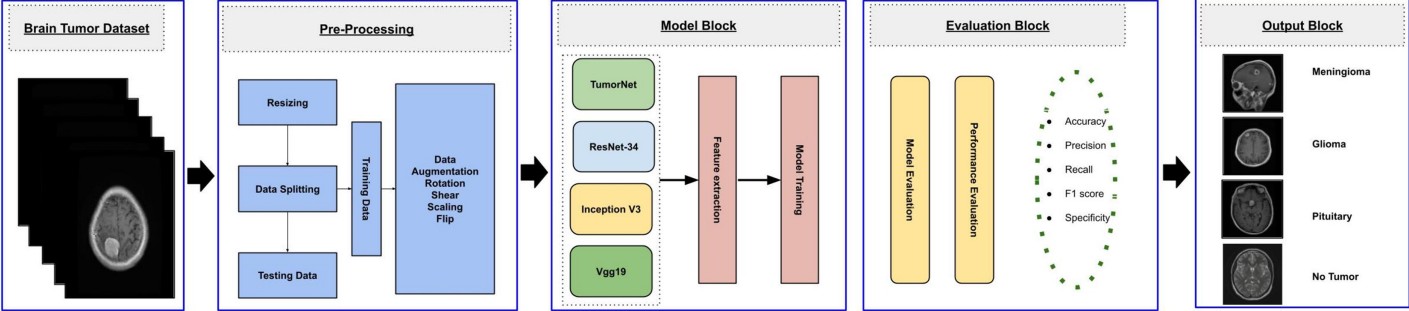

**Figure 1  Visual representation of overall methodology.**               

an accuracy of 99.3%. Recent studies have shown that integrating CNNs with attention mechanisms can significantly enhance feature representation and improve classification accuracy in brain tumor detection using MRI (*Rasheed et al., 2024*). 3D CNNs have also been effectively applied to neuroimaging-based disease classification, where PET and SPECT data were used to distinguish between Alzheimer's and Parkinson's diseases with high accuracy (*Tufail et al., 2021*). An efficient feature selection approach combined with explainable classification was successfully applied to EEG-based epileptic seizure detection, demonstrating the importance of interpretable models in neurological disorder diagnosis (*Ahmad et al., 2024*). The impact of combining Batch Normalization and Dropout in neural networks was explored, highlighting how their interaction can affect early-stage Alzheimer's disease classification performance and model stability (*Tufail et al., 2022*).

## PROPOSED METHODOLOGY

This study aims to develop an efficient CNN-based architecture for accurately classifying various types of brain tumors. The methodology begins with preprocessing the input MRI dataset, which is then divided into training, validation, and testing subsets. The training subset is utilized to train both custom CNN models, built from scratch, and pre-trained transfer learning models. The validation subset is used to monitor the model's performance during training, while the testing subset is employed for final evaluation to ensure robustness and reliability. The workflow of the proposed approach is outlined in Fig. 1.

### Dataset description

This study utilizes a dataset of high-quality, T1-weighted contrast-enhanced MRI scans, consisting of 3,064 samples gathered from two prominent medical institutions in China: Nanfang Hospital in Guangzhou and the General Hospital of Tianjin Medical University. These images were collected over a period from 2005 to 2010, offering a rich and varied set of clinical data. The dataset includes three major brain tumor types: meningiomas, gliomas, and pituitary tumors, all of which are frequently encountered in clinical settings (Figshare) BraiN Tumor Figshare Dataset.

The dataset was originally preprocessed by Figshare for the purpose of building a brain tumor classification model. It contains 2D MRI slices from 233 patients, with patient

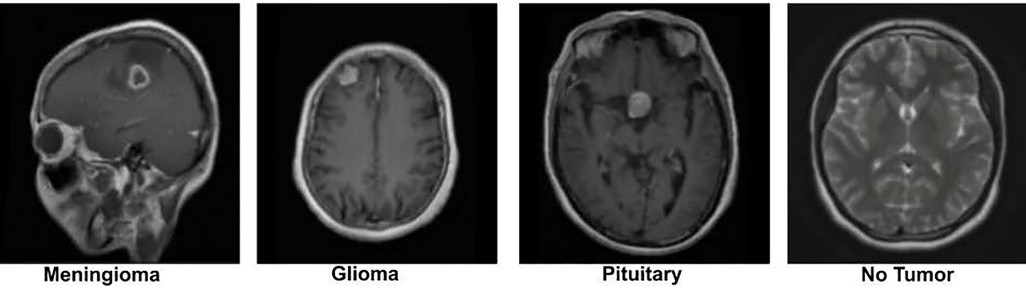

**Figure 2** Visual representation of sample images from the dataset (*Nickparvar*).

identities removed to ensure strict privacy compliance. The data is categorized into three tumor types: the meningioma subset has 708 slices, the glioma subset has 1,426 slices, and the pituitary tumor subset contains 930 slices. These slices are labeled according to the tumor type, which serves as the ground truth for analysis. To further enrich the dataset, we also incorporated additional images from a publicly available Kaggle brain tumor dataset, which contains a "No Tumor" category as well (*Nickparvar*; Kaggle dataset). This addition helps to provide a balanced dataset, including both tumor and non-tumor cases. A sample image from the dataset is displayed in Fig. 2.

The dataset utilized in this study is compiled from three reliable open-access sources: Figshare, the SARTAJ dataset, and Br35H, all available *via* Kaggle. It includes four distinct categories: Pituitary, Meningioma, Glioma, and No Tumor—thereby increasing the total sample size and enhancing class variability. The distribution of these four classes across the combined dataset is depicted in Fig. 3.

## Pre processing

To prepare MRI images from the Figshare dataset for classification (*Tufail et al., 2021*), a well-defined preprocessing pipeline was developed to ensure compatibility with multiple neural network architectures and to support effective model training. The original images were provided in MATLAB's.mat format. These were first processed by expanding their dimensions where necessary, allowing for proper loading and manipulation in Python-based environments. To accommodate different model requirements, two standardized image sizes were used. Images were resized to 224 × 224 pixels for compatibility with pretrained architectures such as VGG19 and ResNet-34, which require this input size. In parallel, images were also resized to 256 × 256 pixels to align with the input shape expected by the custom-designed CNN developed in this study. This dual-resizing approach ensured that all models could be trained and evaluated consistently without compromising structural integrity. Following resizing, all images were converted into NumPy arrays to reduce storage overhead and facilitate efficient data handling during training. The full dataset was then randomized to eliminate any order-based bias and was stratified into training, validation, and testing sets using a 70:10:20 split. This careful allocation aimed to maintain class balance across subsets, promoting a reliable evaluation of model performance while minimizing the risk of overfitting. This structured

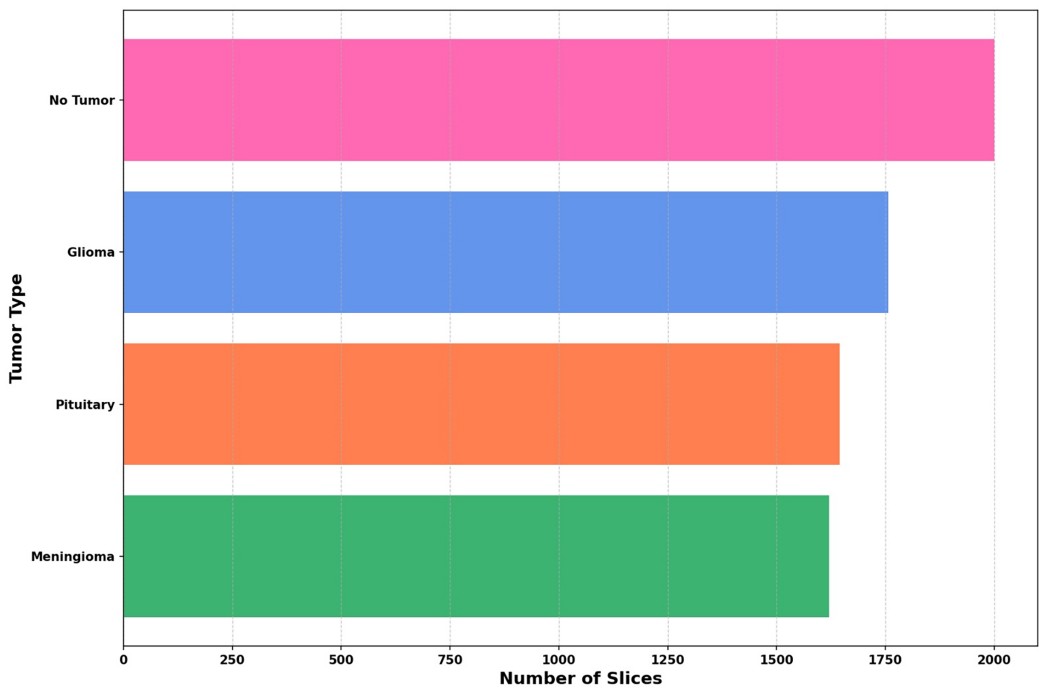

**Figure 3** The distribution of images across each class is illustrated in a bar graph.

preprocessing and data splitting strategy played a critical role in enhancing the reproducibility, reliability, and generalizability of the classification results.

## Data augmentation

To address the challenge of limited training data, data augmentation was employed. This technique expands the dataset by applying transformations such as brightness adjustment, scaling, flipping, random cropping, zooming, and contrast adjustment to existing images, generating new variations while preserving their original labels. By effectively increasing the dataset size, data augmentation enables deep learning models to leverage more diverse data, improving training performance and robustness (*LeCun et al., 2002*). Data augmentation also serves as a form of dataset-level regularization, mitigating overfitting by introducing variability into the training data. This approach enhances the model's ability to generalize to unseen data without requiring changes to the model architecture. Additionally, data augmentation addresses the problem of class imbalance by oversampling minority classes, creating a more balanced training dataset, and ensuring unbiased learning (*Shorten & Khoshgoftaar, 2019*). Medical imaging datasets are often small and challenging to collect due to the complexity of data acquisition and labeling. Data augmentation has been shown to be highly effective in enhancing performance in tasks such as skin lesion classification, liver lesion detection, and brain scan analysis. In this study, data augmentation increased the dataset to a total of 11,200 images, with 2,800 images per class. Figure 4 illustrates the augmentation techniques applied, showcasing the

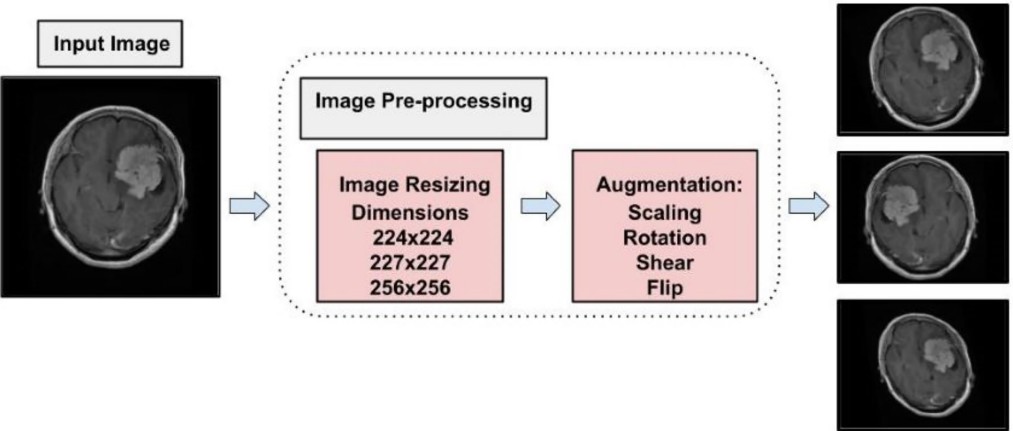

**Figure 4 Visual representation of augmentation process.**

diversity introduced into the training dataset. This strategy significantly improved model generalization, diagnostic accuracy, and robustness.

Figure 5 illustrates the distribution of the dataset for each class after augmentation. This visualization highlights how the data has been expanded to ensure a more balanced representation of each class. The augmentation process helps improve the model's ability to generalize by introducing variations in the training data, which are critical for training more robust models.

The dataset is segmented into three phases: the first phase involves model training, where the model learns from the available data; the second phase is dedicated to model evaluation, which focuses on hyperparameter tuning and performance optimization; and the final phase is the test dataset, used to assess the model's ability to generalize to new, unseen data. The distribution of the dataset across these phases is detailed in Table 1.

## Leveraging transfer learning for feature extraction

Transfer learning is a powerful technique that eliminates the need to train models from scratch for each dataset by utilizing knowledge embedded in pre-trained models (*Tan et al., 2018*). By adapting these models to new data, transfer learning minimizes the need for extensive training, reducing time, computational resources, and costs while enhancing performance (*Pan & Yang, 2009*). This approach is particularly advantageous in scenarios where labeled data is scarce or expensive to obtain and is graphically represented in Fig. 6. In this study, we employed four widely recognized and parameterized pre-trained CNN for feature extraction. These models were carefully chosen based on their established relevance, robust architecture, and frequent application in the literature, ensuring the reliability of our results and facilitating meaningful comparisons with related studies. By fine-tuning these pre-trained models on our dataset, we effectively harnessed their pre-learned feature representations, enabling improved performance, generalization, and computational efficiency.

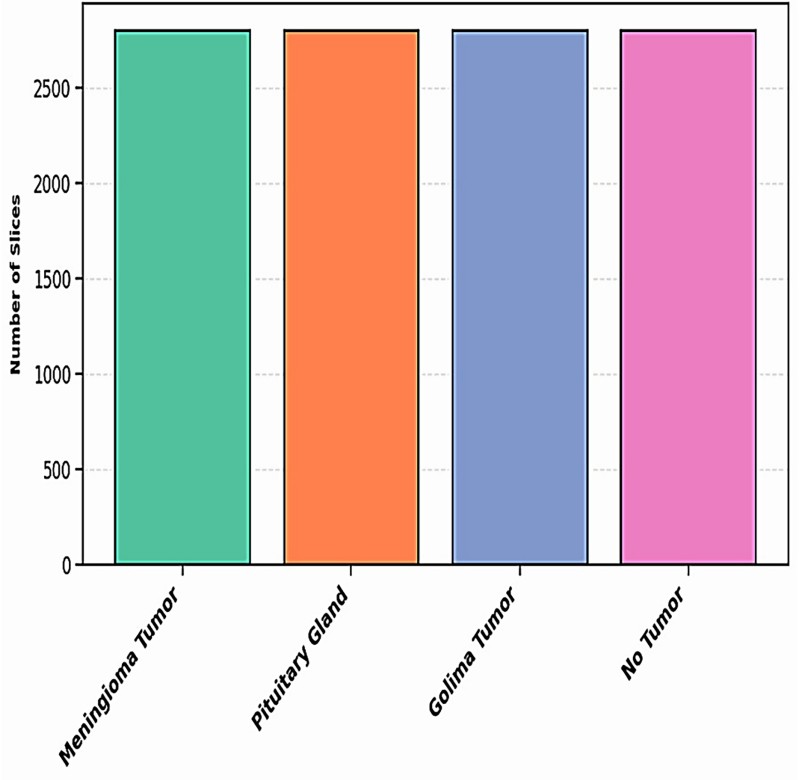

**Figure 5** **The distribution of dataset after augmentation.**

**Table 1 Dataset distribution across categories.**

| Category | Number of images |
| --- | --- |
| Training | 7,840 |
| Validation | 2,240 |
| Test | 1,120 |

## VGG19

The VGG19 model, also known as VGGNet-19, is an extension of the VGG16 architecture, featuring 19 convolutional layers instead of 16. Developed by Karen Simonyan and Andrew Zisserman in 2014, VGG19 was trained on the ImageNet dataset, a large-scale visual database widely used in object recognition research. Despite its increased depth, VGG19 retains the simplicity and uniformity of the VGG family, using small convolutional filters (3 × 3) throughout its architecture. This design emphasizes deeper network structures to capture more complex features, making it a robust choice for feature extraction in various applications (*Simonyan & Zisserman, 2014*). Figure 7 illustrates the detailed architecture of VGG19.

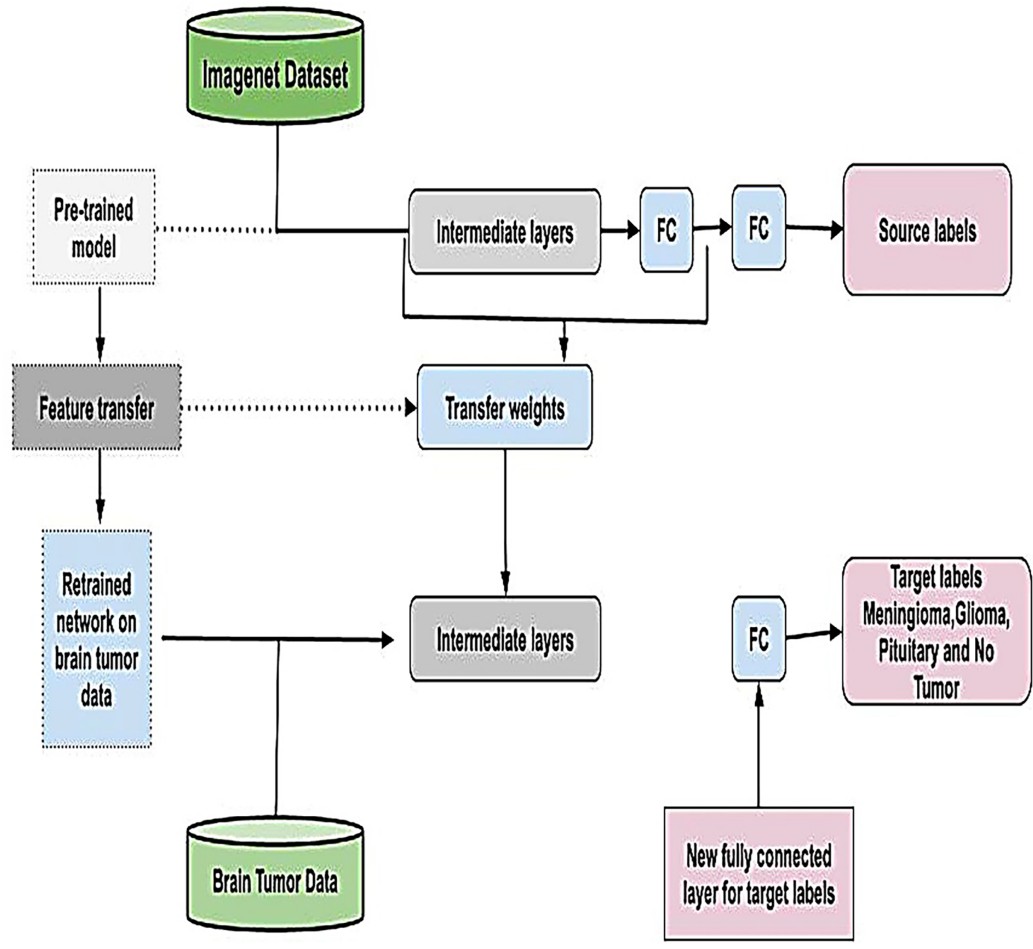

**Figure 6 Visual representation of transfer learning approach.**

## InceptionV3

InceptionV3 is a widely recognized pre-trained CNN renowned for its innovative architecture, which balances accuracy and computational efficiency. Its design incorporates a combination of convolutional layers with various kernel sizes $1 \times 1$, $3 \times 3$, and $5 \times 5$ alongside max pooling and average pooling layers. This multi-scale approach allows the model to extract features of varying complexity, enabling it to capture intricate patterns in images with remarkable precision (*Islam et al., 2022*). To further enhance performance, InceptionV3 integrates advanced techniques such as batch normalization, which stabilizes and accelerates training, and dropout, which prevents overfitting by regularizing the network. Additionally, the use of factorized convolutions reduces computational overhead while maintaining the model's ability to learn rich feature representations. Auxiliary classifiers embedded within the network facilitate gradient flow during training, improving both convergence and accuracy. These features collectively contribute to InceptionV3 state-of-the-art performance on numerous image recognition benchmarks. To simplify its understanding, the InceptionV3 architecture can be broken

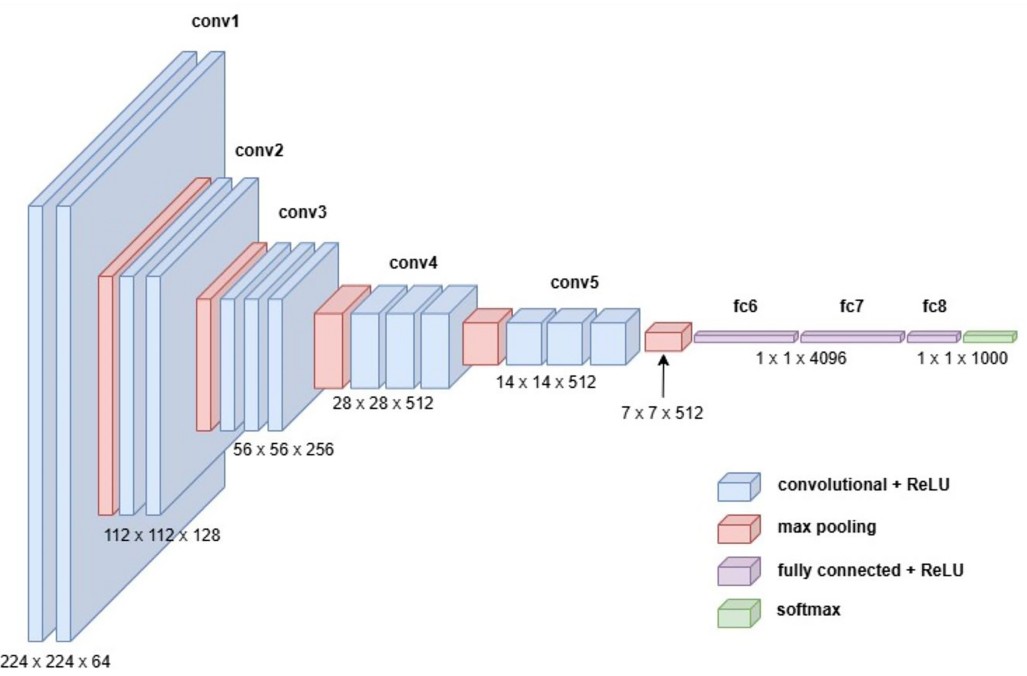

**Figure 7 Visual representation of VGG-19 architecture.**

down into modular components, with repeated layers denoted as x2, x3, or x4 where applicable. This segmentation highlights the network's systematic organization and its ability to scale efficiently. Figure 8 provides a detailed visual representation of the architectural layout, illustrating the interplay of its innovative components.

## ResNet-34

ResNet-34, a more extensive model compared to ResNet-18, was introduced by *He et al. (2016)*. It follows the residual learning framework, utilizing skip or identity connections that bypass certain layers to prevent the vanishing gradient problem, enabling the training of deeper networks. With 34 layers, ResNet-34 strikes a balance between complexity and performance. During benchmark tests, it played a crucial role in the success of the ResNet ensemble, notably in the ILSVRC 2015 competition. ResNet-34 has demonstrated its robustness across several vision tasks, such as image classification, detection, and segmentation (*He et al., 2016*). Figure 9 provides a detailed visual representation of ResNet-34 architecture.

## Proposed models

To develop an optimal architecture for classifying brain tumors from MRI images, we conducted a series of structured experiments to understand how CNNs extract and interpret features from the input data. CNNs are composed of multiple layers that identify increasingly abstract representations, starting with low-level features such as edges and textures, and progressing to complex structures critical for accurate classification. In each convolutional layer, learnable filters are applied to extract spatial features, resulting in the

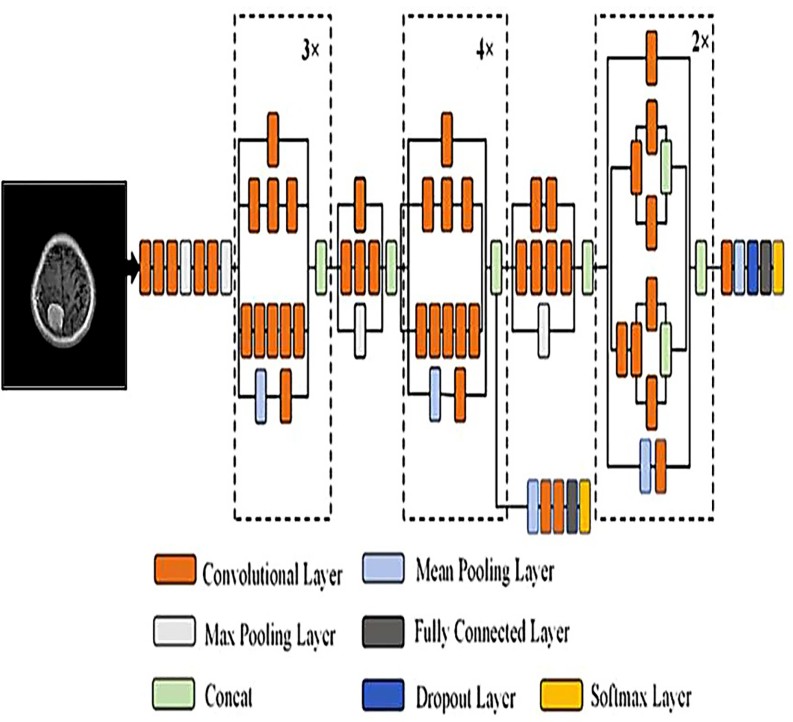

**Figure 8 Visual representation of Inception V3 architecture.**

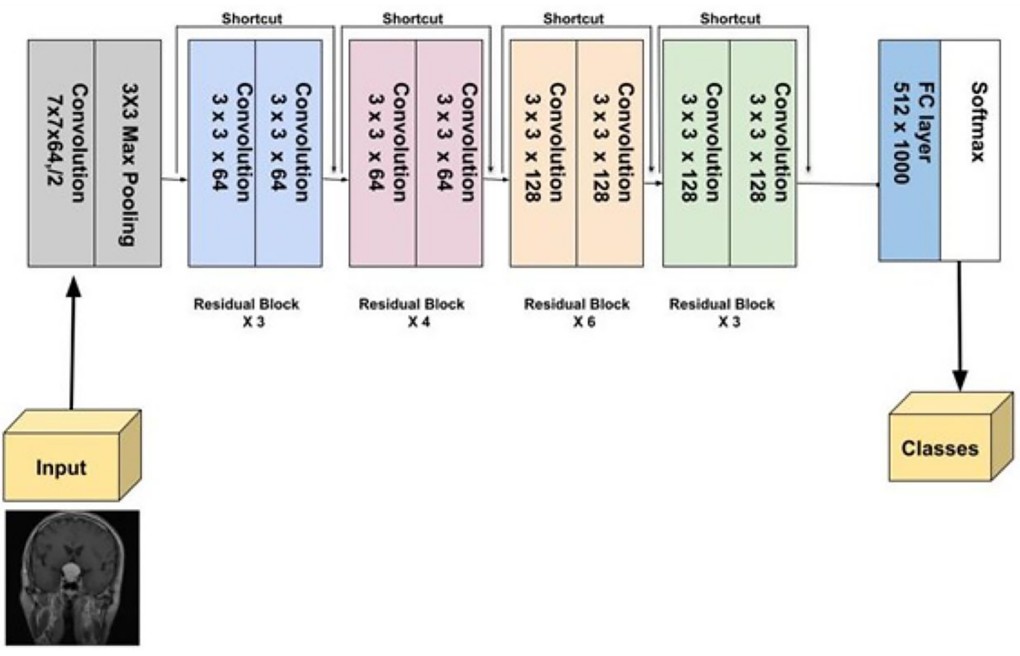

**Figure 9 Visual representation of ResNet-34.**

generation of feature maps that emphasize relevant image patterns. Parameters such as stride and padding control how these filters move across the input, directly influencing the resolution of the output feature maps. Pooling layers, often used after convolution, downsample the feature maps to reduce computational cost while preserving the most significant information. To enhance non-linear learning capabilities, activation functions like ReLU are incorporated, enabling the network to model complex relationships inherent in medical imaging data. This study introduces and evaluates three custom CNN architectures BrainNet-1, BrainNet-2, and BrainNet-3 each progressively refined to enhance accuracy, generalization, and computational efficiency. The initial model, BrainNet-1, served as a baseline and comprised a conventional stack of convolutional and pooling layers, followed by fully connected layers. This architecture established a reference point for subsequent improvements. Building upon this foundation, BrainNet-2 integrated deeper layers, varying kernel sizes, and dropout regularization to reduce overfitting— particularly effective in handling the variability seen in MRI datasets. The model also emphasized preserving spatial resolution to retain rich feature representations. The final model, BrainNet-3, was tailored specifically for the unique challenges of MRI-based brain tumor classification, including high intra-class variability and data imbalance. This architecture employed a carefully optimized combination of convolutional and pooling blocks, and incorporated attention mechanisms to enhance the network's focus on diagnostically relevant features. As a result, BrainNet-3 achieved notable improvements in classification precision, recall, and overall robustness. These three custom-designed architectures showed consistent performance gains across development stages. The task-specific design of the BrainNet series allowed them to outperform several widely used pre-trained models, reinforcing the advantage of domain-focused architectural optimization. Detailed architectural configurations for BrainNet-1, 2, and 3 are illustrated in Figs. 10, 11, and 12, respectively. Additionally, the hyperparameters used for training each model were carefully selected to achieve an optimal balance between computational complexity and predictive accuracy. To evaluate the architectural efficiency and complexity of our proposed models, we compared three variants BrainNet-1, BrainNet-2, and BrainNet-3 in terms of total trainable parameters. As shown in Table 2, BrainNet-1 contains the highest number of parameters due to its large fully connected layer following a shallow convolutional structure. In contrast, BrainNet-2 and BrainNet-3 employ deeper convolutional layers with progressive spatial reduction, resulting in significantly fewer parameters. BrainNet-3, the most compact of the three, achieves a balance between depth and computational efficiency, making it suitable for deployment in resource-constrained environments. The hyperparameter values are summarized in Table 3.

## RESULTS AND DISCUSSION

To evaluate the performance and efficiency of our system, we utilized a comprehensive set of metrics, including precision, recall, F1-score, support, and accuracy (*Bishop, 2006*). Since the dataset used in this study is balanced, accuracy provides a reliable measure of overall model performance. However, to ensure a thorough evaluation, we also incorporated precision, which measures the proportion of correctly predicted positive

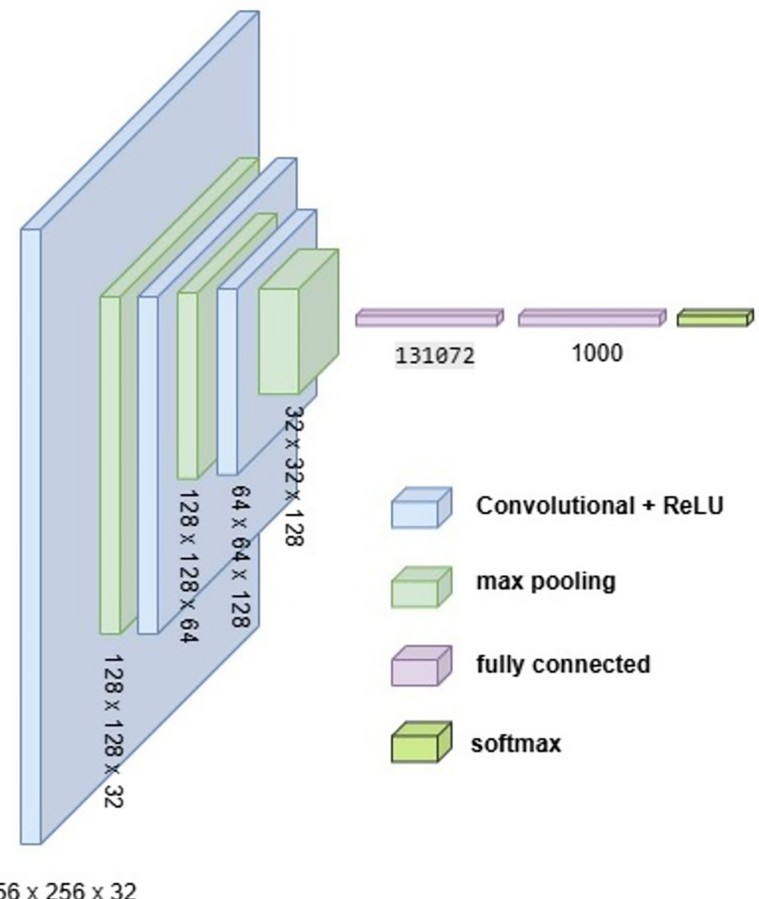

**Figure 10 Visual representation of BrainNet-1 architecture.**

cases, and recall, which assesses the model's ability to identify all relevant instances. The F1-score, as the harmonic mean of precision and recall, provided a single metric that balances these two aspects. Support further contextualized the evaluation by reflecting the number of instances in each class, offering deeper insights into classification effectiveness.

The model training was performed on an HP Victus system equipped with an Intel Core i7 processor, 32 GB RAM, and an NVIDIA GeForce RTX 4060 GPU with 8 GB VRAM. The implementation was conducted in Python 3.6 using the PyTorch framework, known for its computational efficiency and flexibility. The average execution time per epoch was approximately 30 s, highlighting the system's ability to efficiently handle the training process. The subsequent section provides a detailed analysis of the results obtained from the self-developed models, comparing them with transfer learning approaches and discussing their contributions to classification performance.

## Evaluating the performance of proposed methods

The classification report is a critical tool for evaluating machine learning algorithms in classification tasks. It includes key metrics such as accuracy, precision, recall, F1-score

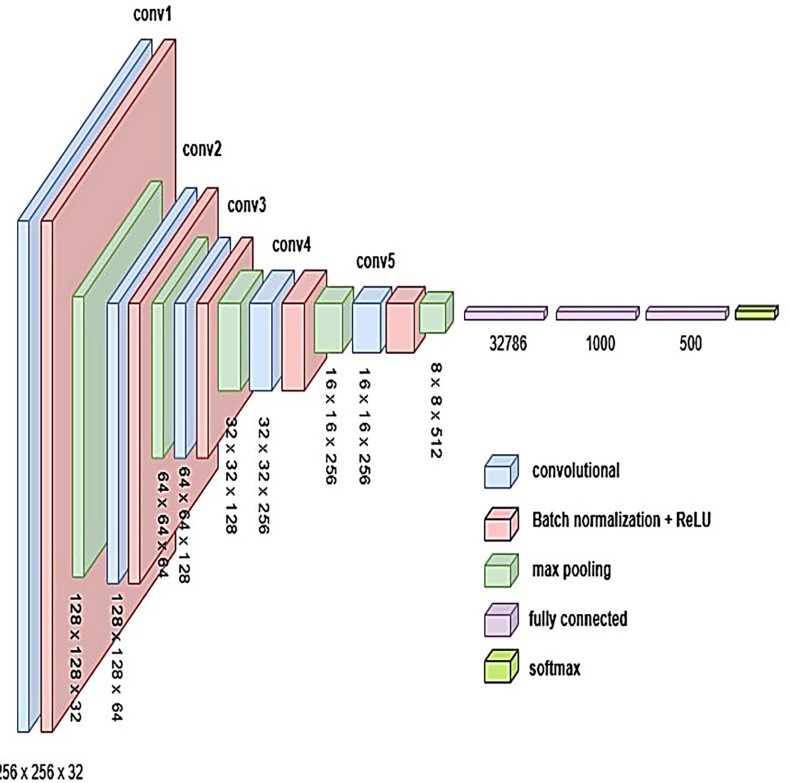

**Figure 11 Visual representation of BrainNet-2 architecture.**

(*Bishop, 2006*). While some of these metrics offer a high-level overview of model performance, others provide deeper, more detailed insights into specific aspects of the model's behavior. This comprehensive evaluation enables a more nuanced understanding of the model's effectiveness, facilitating informed decisions regarding model selection and deployment. For completeness, we provide the mathematical definitions of these metrics, where true positive (TP), true negative (TN), false positive (FP), and false negative (FN) (*Ferrari, Gonzalez & Schmidt, 2017*) are used to quantify the outcomes of predictions.

## Assessment of predictive methods

In our study, we explored two primary methodologies for classifying brain tumors using MRI images: transfer learning with pre-trained models and the development of custom CNNs. The transfer learning approach involved fine-tuning existing models by modifying their final layers to adapt to our specific dataset, including the addition of fully connected and hidden layers tailored to the classification of tumor types. Alternatively, we designed a custom CNN architecture optimized for our dataset, which demonstrated superior performance in terms of accuracy, precision, and inference speed compared to the fine-tuned pre-trained models. By integrating the rapid feature extraction capabilities of pre-trained models with the adaptability of custom architectures, we leveraged the strengths of both approaches to enhance classification effectiveness. This dual strategy

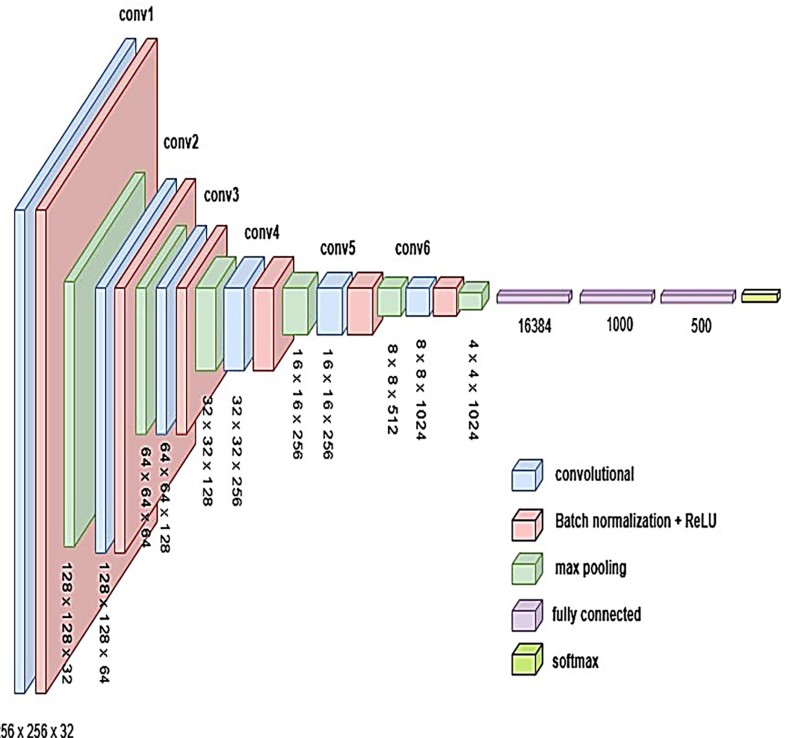

**Figure 12 Visual representation of BrainNet-3 architecture.**

**Table 2 Total trainable parameters of the proposed models.**

| Model | Total parameters (Millions) |
| --- | --- |
| BrainNet-1 | 131.32 |
| BrainNet-2 | 35.43 |
| BrainNet-3 | 23.76 |

**Table 3 Optimized hyper-parameters used in the model.**

| Hyper-parameters | Optimized value |
| --- | --- |
| Optimizer used | Adam |
| Epoch | 40 |
| Learning rate | 0.001 |
| Batch size | 256 |
| Criterion | Cross entropy |

provided a comprehensive baseline for patient management and informed decision-making. Each methodology was rigorously evaluated to ensure a thorough assessment of performance metrics, thereby validating the effectiveness of our classification strategies. The results underscore the importance of selecting appropriate modeling techniques based on dataset characteristics and specific clinical objectives.

## Evaluation with transfer learning

In this research, we assessed the performance of three established CNN architectures, ResNet-34 (*He et al., 2016*), VGG19 (*Simonyan & Zisserman, 2014*), and Inception-V3 (*Islam et al., 2022*) by retraining them on a brain tumor dataset containing four distinct categories: Glioma, Meningioma, Pituitary tumor, and No tumor. The large and diverse nature of this dataset provided a solid basis for both the training and evaluation of the models, enabling a comprehensive analysis of their ability to classify medical images accurately.

### Training and validation

The dataset was partitioned into training and validation sets to facilitate the model development process. The models were trained using the cross-entropy loss function, which is a standard approach for multi-class classification problems. The training set was used to adjust the model parameters, while the validation set was utilized to periodically assess the models' performance on unseen data during training. This allowed for monitoring overfitting and ensured that the models did not become too tailored to the training data. To further mitigate overfitting and ensure robust performance, early stopping and model checkpointing strategies were implemented. Early stopping was applied to halt training once the model's performance on the validation set ceased to improve for a specified number of epochs. This prevented unnecessary computation and avoided overfitting. Additionally, data augmentation techniques were employed to artificially increase the diversity of the training data. This involved applying random transformations to the input images, such as rotation, flipping, and scaling, which enabled the models to better generalize to new, unseen images. The loss *vs.* epoch graph for all three models Fig. 9 provides a visual representation of the training process. The graph demonstrates how the loss for each model decreases steadily over time, with both training and validation losses following similar trends. The ResNet-34 model showed the most rapid decrease in loss, achieving faster convergence, while Inception-V3 and VGG19 also exhibited consistent improvements, though the rate of convergence was slightly slower in comparison. These results indicate the effectiveness of the training strategies applied and the ability of the models to learn from the provided dataset.

The accuracy and loss of each model on the training and validation datasets over time is depicted in Figs. 13 and 14. As shown in the Fig. 14, the accuracy of all models increased steadily during the training process, with validation accuracy mirroring the trend observed in the training data. This steady improvement suggests that the models were effectively learning from the training data and generalizing well to unseen examples in the validation set. The increase in validation accuracy over time further indicates that the models were not overfitting to the training data, as the validation accuracy did not plateau prematurely or decrease over time. Instead, the consistent rise in accuracy for both training and validation sets demonstrates the successful application of training strategies and the models' ability to generalize effectively. These results highlight the models' robustness and suggest that the training process was well-optimized, with no signs of overfitting or

underfitting throughout the training period. Table 4 shows the comparison of training and validation metrics for different models.

## Model evaluation

After training, the models were evaluated on the test set to determine their final performance. ResNet-34 achieved an accuracy of 97.38%, Inception-V3 reached 97.02%, and VGG19 attained 96.19%. These results demonstrate the potential of deep learning models in medical image classification, particularly when trained on large, diverse datasets. The high accuracy rates suggest that the models were successful in learning discriminative features from the data and were able to generalize well to new images. These findings highlight the effectiveness of deep learning models in medical image analysis. However, it is important to note that the creation of large datasets for training can be challenging in some contexts. In such situations, techniques such as transfer learning, cross-validation, and data augmentation can be employed to maximize the utility of available data. Furthermore, conducting multiple experiments and selecting the most optimal model based on performance metrics can further enhance classification outcomes. These strategies are essential for building reliable predictive models, especially when data resources are limited. Table 5 displays the test accuracy for all the transfer learning models.

## Evaluation of custom-built CNN

In this study, we developed and evaluated three custom CNN architectures, BrainNet-1, BrainNet-2, and BrainNet-3 designed specifically for classifying brain tumors. The models were trained using two distinct datasets: the FigShare dataset (*Tufail et al., 2021*) and a Kaggle brain tumor dataset (*Ahmad et al., 2024*).

- FigShare dataset: This dataset comprises 3,064 T1-weighted contrast-enhanced MRI images and is categorized into three tumor classes: meningioma, glioma, and pituitary tumors.
- Kaggle dataset: In addition to the FigShare dataset, we also used the Kaggle brain tumor dataset, which consists of MRI images for four classes: glioma, meningioma, pituitary, and no tumor. This dataset significantly expanded the variety of images used for training, ensuring that the models were exposed to a broader spectrum of tumor types, as well as healthy brain scans.

Both datasets were carefully preprocessed to ensure consistency, including normalization of image pixel values, resizing to a standard dimension, and augmenting the data to improve model robustness.

## Training and validation

The combined datasets were split into training and validation sets to facilitate the model development process. The training set was used to adjust the model parameters, while the validation set served to periodically evaluate the models' performance during training, helping to monitor overfitting. The models were optimized using the cross-entropy loss function, a standard approach for multi-class classification tasks, and the Adam optimizer

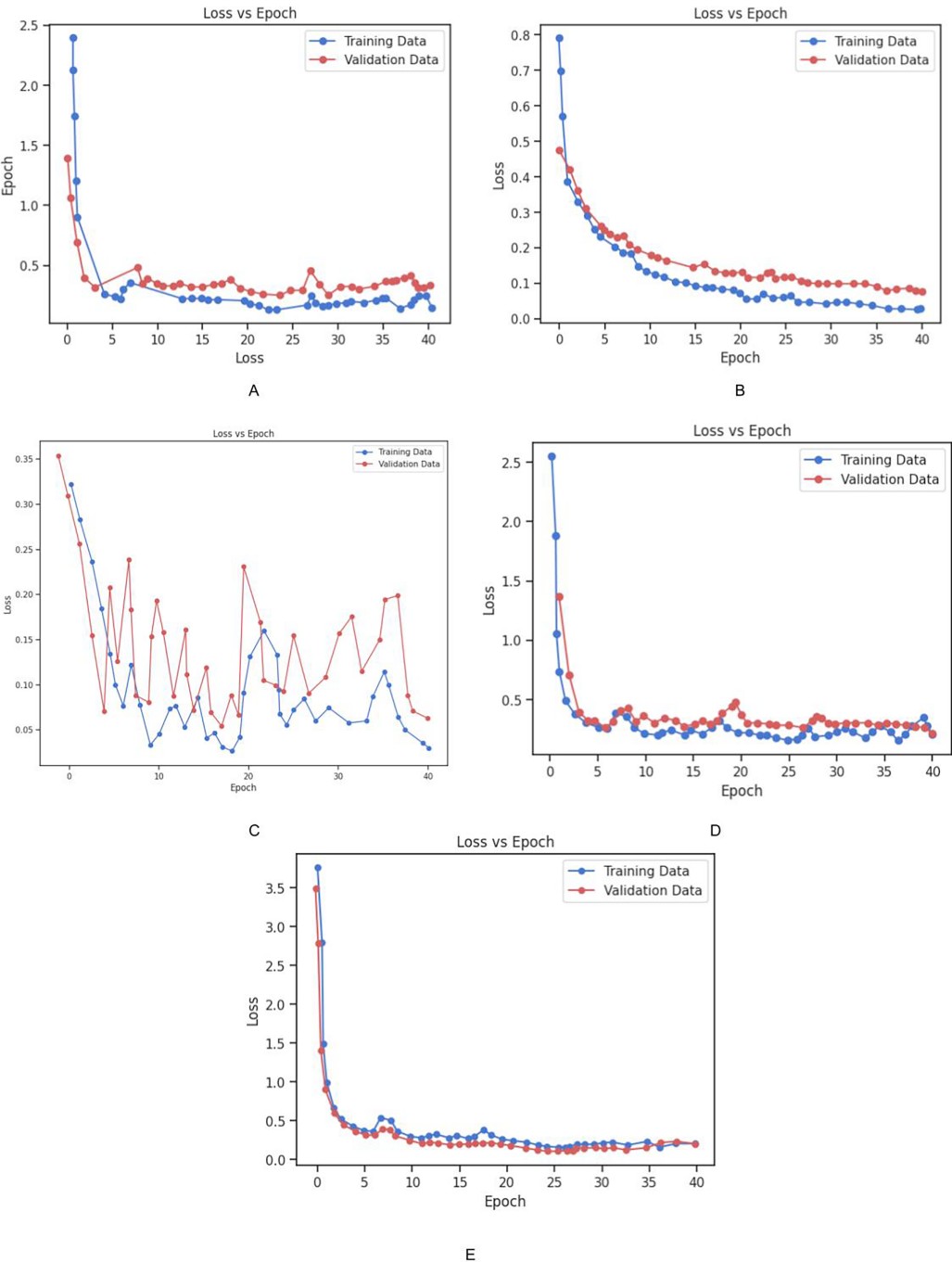

**Figure 13 Graphical representation of Loss *vs*. Epoch for all the models.** (A) Training and validation loss for Inception_v3 Model. (B) Training and validation loss for Vgg_19 Model. (C) Training and validation loss for BrainNet1 Model. (D) Training and validation loss for BrainNet2. (E) Training and validation loss for BrainNet3 Model.

was employed for efficient convergence. To mitigate overfitting and ensure robust performance, several regularization techniques were applied. Data augmentation was used to artificially expand the training set by applying random transformations to the images,

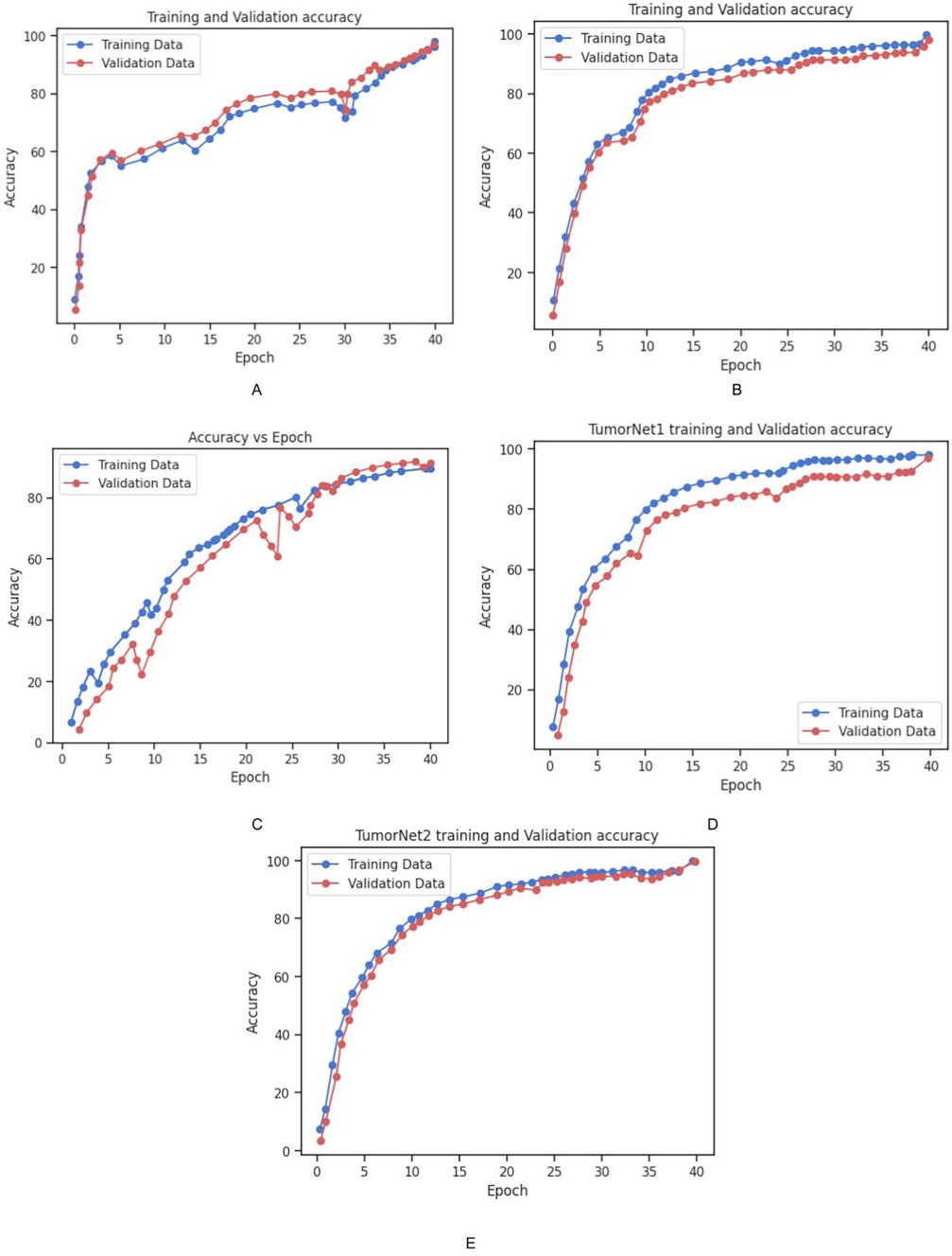

**Figure 14 Graphical representation of Accuracy *vs.* Epoch for all the models.** (A) Training and validation accuracy for Inception_v3 Model. (B) Training and validation accuracy for Vgg_19 Model. (C) Training and validation accuracy for BrainNet1 Model. (D) Training and validation accuracy for BrainNet2. (E) Training and validation accuracy for BrainNet3 Model.

such as rotation, flipping, zooming, and shifting. These augmentations increased the variability of the training data, helping the models generalize better to unseen images. Furthermore, early stopping was utilized to halt the training process once the validation

**Table 4 Comparison of training and validation metrics for different models.**

| Model | Training accuracy (%) | Training loss | Validation accuracy (%) | Validation loss |
| --- | --- | --- | --- | --- |
| Inception_V3 | 98.02 | 0.14 | 97.02 | 0.33 |
| Vgg19 | 99.61 | 0.02 | 98.07 | 0.07 |
| Resnet-34 | 98.12 | 0.014 | 98.81 | 0.02 |

**Table 5 Test accuracy comparison of different models.**

| Model | Test accuracy (%) |
| --- | --- |
| Vgg19 | 96.89 |
| Inception V3 | 97.02 |
| ResNet 34 | 97.38 |

accuracy no longer improved for a specified number of epochs, thus preventing unnecessary training and overfitting. Model checkpointing was also implemented to save the model with the best validation performance during the training process, ensuring that the optimal weights were preserved. The training and validation loss for all models are shown in Fig. 13. The graph illustrates the convergence of the models during training, with both training loss and validation loss steadily decreasing as the models learned from the data. This behavior indicates that the models were able to effectively capture the patterns in the dataset while avoiding overfitting.

Additionally, the training and validation accuracy for each model is presented in Fig. 14. The graph shows a steady increase in training accuracy for all models, with the validation accuracy following a similar trend. The models consistently improved their performance over time, with validation accuracy increasing as the models learned to generalize from the training data. This indicates that the models were effectively trained, and their performance did not degrade with additional epochs, further demonstrating their robustness and ability to generalize to unseen data. Table 6 represents the training and vaidation performance metrics for all the proposed models.

### Evaluation

The performance of the three BrainNet models was assessed on both the training and validation datasets using several key metrics, including accuracy, precision, recall, and F1-score. These models were trained to identify and classify brain tumor types, and the evaluation results revealed significant differences in performance across the architectures.

- BrainNet-1 achieved an accuracy of 94.94%.
- BrainNet-2 improved to 98.98%
- BrainNet-3 reached an impressive 99.92%

These results underscore the influence of model complexity on performance, with the more complex architectures (BrainNet-2 and BrainNet-3) demonstrating superior accuracy compared to the simpler BrainNet-1. Furthermore, BrainNet-3 provided the best

**Table 6 Training and validation performance metrics for BrainNet models.**

| Model | Training accuracy (%) | Training loss | Validation accuracy (%) | Validation loss |
|---|---|---|---|---|
| BrainNet 1 | 94.55 | 0.40 | 95.81 | 0.32 |
| BrainNet 2 | 98.01 | 0.20 | 97.02 | 0.21 |
| BrainNet 3 | 99.02 | 0.20 | 98.01 | 0.20 |

balance between accuracy and computational efficiency, making it the most suitable model for classifying brain tumor MRI images. The Table 6 presents training and validation performance metrics for BrainNet models. Figure 13 shows the Graphical representation of Loss *vs.* Epoch for all the models while Fig. 14 shows the Graphical representation of Accuracy *vs.* Epoch for all the models. Table 7 represents the test accuracies of all the proposed models adnd Table 8 represents the performance metrics (precision, recall, F1-score, and accuracy) for proposed models and transfer learning models.

# COMPARISON WITH STATE OF THE ART

In this research, we developed a deep learning model, TumorNet, to classify brain tumors using MRI images. We trained the model on a benchmark dataset comprising of four classes. To enhance the dataset and improve classification accuracy, we applied various data augmentation techniques, including horizontal and vertical flipping, rotation, transposition, zooming, ZCA whitening, shearing, and brightness adjustments. Our methodology involved utilizing BrainNet alongside established models such as Inception-V3 (*Islam et al., 2022*), ResNet-34 (*He et al., 2016*), and VGG19 (*Simonyan & Zisserman, 2014*). ResNet-34 (*He et al., 2016*), part of the Residual Networks family, incorporates shortcut connections to bypass one or more layers, effectively addressing issues of degrading accuracy and vanishing gradients in deep networks. We modified the fully connected layer of ResNet-34 (*He et al., 2016*) to include three neurons, corresponding to our class count, and fine-tuned the model on our dataset. We evaluated the performance of our model using multiple metrics: accuracy, precision, recall, F1-score, and balanced accuracy. While accuracy is a commonly used performance measure, it can be misleading with imbalanced datasets, as it may favor the largest class. Therefore, we also calculated precision, recall, F1-score, and balanced accuracy to provide a more comprehensive evaluation of the model's performance. Comparing our results with previous studies on the same benchmark dataset, we observed significant improvements. Earlier works achieved accuracies ranging from 84.19% to 93.68%. Our proposed model outperformed these, achieving a higher accuracy of 99.88%, surpassing the best previous work. By utilizing transfer learning, the ResNet-34 architecture demonstrated its ability to effectively differentiate MRI images, overcoming challenges like overlapping intensities, tumor size variations, and low contrast. Its deeper layers allowed the network to learn more abstract features, making it particularly well-suited for handling heterogeneous MRI data. Similarly, InceptionV3 (*Islam et al., 2022*) and VGG19 (*Simonyan & Zisserman, 2014*) also showcased strong performance, leveraging their unique architectural strengths. InceptionV3's ability to capture multiscale features and VGG19's deeper layers both

**Table 7 Test accuracy of BrainNet models.**

| Model | Test accuracy (%) |
|---|---|
| BrainNet 1 | 94.94 |
| BrainNet 2 | 98.98 |
| BrainNet 3 | 99.92 |

**Table 8 Performance metrics (Precision, Recall, F1-score, and Accuracy) for various models and classes.**

| Model | Class | Precision | Recall | F1-score | Accuracy% |
|---|---|---|---|---|---|
| ResNet-34 | Meningioma | 0.96 | 1.00 | 0.98 | 98.50 |
|  | Glioma | 0.99 | 0.96 | 0.97 |  |
|  | Pituitary | 1.00 | 0.98 | 0.99 |  |
|  | No Tumor | 1.00 | 1.00 | 1.00 |  |
| Vgg-19 | Meningioma | 0.95 | 1.00 | 0.97 | 97.75 |
|  | Glioma | 0.97 | 0.97 | 0.97 |  |
|  | Pituitary | 0.99 | 0.94 | 0.97 |  |
|  | No Tumor | 1.00 | 1.00 | 1.00 |  |
| Inception-v3 | Meningioma | 0.97 | 1.00 | 0.98 | 98.02 |
|  | Glioma | 0.97 | 0.96 | 0.96 |  |
|  | Pituitary | 0.99 | 0.96 | 0.97 |  |
|  | No Tumor | 1.00 | 1.00 | 1.00 |  |
| BrainNet-1 | Meningioma | 0.92 | 0.94 | 0.93 | 94.94 |
|  | Glioma | 0.91 | 0.95 | 0.93 |  |
|  | Pituitary | 0.97 | 0.94 | 0.95 |  |
|  | No Tumor | 1.00 | 0.96 | 0.98 |  |
| BrainNet-2 | Meningioma | 0.98 | 1.00 | 0.99 | 98.98 |
|  | Glioma | 0.98 | 0.98 | 0.98 |  |
|  | Pituitary | 1.00 | 0.99 | 0.99 |  |
|  | No Tumor | 1.00 | 0.99 | 0.99 |  |
| BrainNet-3 | Meningioma | 1.00 | 1.00 | 1.00 | 99.92 |
|  | Glioma | 1.00 | 1.00 | 1.00 |  |
|  | Pituitary | 1.00 | 1.00 | 1.00 |  |
|  | No Tumor | 1.00 | 1.00 | 1.00 |  |

contributed to improved accuracy. Additionally, data augmentation techniques were employed to enrich the dataset, providing more varied inputs that further enhanced model performance and robustness. Earlier methods, particularly those based on classical machine learning, involved manual feature extraction, which is time-consuming and resource-intensive. Other studies utilizing CNN architectures employed shallower structures, limiting the number of non-linear layers and restricting high-level feature learning, thereby affecting accuracy. The selection of the ResNet architecture for brain tumor classification proved successful, with the model performing on par with state-of-the-art models. For future work, we plan to expand this study by conducting experiments

**Table 9 Model accuracies and datasets used in various studies.**

| Source | Model | Dataset used | Accuracies (%) |
|---|---|---|---|
| *Kumar & Mankame (2020)* | CNN | SimBRATS | 96.3 |
| *Sharif et al. (2022)* | ANN | BRATS 2018 | 99.0 |
| *Amin et al. (2020)* | SVM | Harvard | 97.1 |
| *Woźniak, Siłka & Wieczorek (2023)* | Corr learning mechanism | Kaggle | 96.0 |
| *Rajat Mehrotra, Agrawal & Anand (2020)* | AlexNet | TCIA | 99.04 |
| *Raja (2020)* | Regression | BRATS 2015 | 98.5 |
| *Tandel et al. (2023)* | BTC_FCNN | FigShare | 98.86 |
| *Cinar, Kaya & Kaya (2023)* | EfficientNet | FigShare | 98.86 |
| *El-Wahab et al. (2023)* | Vgg_16 | FigShare | 98.93 |
| *Zulfiqar, Bajwa & Mehmood (2023)* | ResNet, AlexNet, Vgg_16 | FigShare | 99.30 |
| *Islam et al. (2023)* | InceptionV3 | FigShare, Br35H and Sartaj | 99.60 |
| Proposed | TumorNet, ResNet, AlexNet, Vgg9 | FigShare, Br35H and Sartaj | 99.92 |

with a larger database of brain tumor patients to further improve accuracy. Additionally, we aim to extend the proposed model to other types of medical images, such as radiography (X-ray), ultrasonography, endoscopic, dermoscopy, and histology images. Accurate detection of medical images is crucial due to their high degree of heterogeneity. In this study, we utilized MRI and CT scan images for brain tumor detection. MRI is particularly helpful in diagnosing and staging brain tumors. We applied transfer learning models for brain tumor detection due to their ability to accurately predict tumor cells. Our proposed model, TumorNet, achieved the highest accuracy of 98.81%, and other models also showed improved accuracy with the application of transfer learning techniques. Table 9 presents a comparative analysis of our proposed approach against existing state-of-the-art methods.

## CONCLUSION

In this study, we designed and evaluated a deep learning-based model, BrainNet, for the classification of brain tumors into four distinct categories using MRI images. The model architecture was enhanced by integrating components from established CNN frameworks such as Inception-V3, ResNet-34, and VGG19. To improve model generalization and robustness, various data augmentation techniques were applied, including image flipping, rotation, shifting, zooming, ZCA whitening, shearing, and brightness adjustment. This strategy helped increase the diversity of the training data and reduced the risk of overfitting. Among the tested models, ResNet-34 demonstrated particularly strong performance due to its deep structure and residual connections, which effectively captured complex features in MRI scans. VGG19, known for its simplicity and depth, provided a strong baseline, emphasizing the adaptability of CNN architectures for medical image classification. The proposed model achieved a test accuracy of 98.81%, outperforming several existing methods on the same benchmark dataset. The evaluation was conducted using multiple performance metrics, including accuracy, precision, recall, F1-score, and balanced accuracy, ensuring a comprehensive assessment of model behavior across all

tumor classes. These results highlight the strong potential of deep learning techniques in medical diagnostics. By combining well-established architectures with a custom-designed approach, BrainNet has demonstrated high accuracy and practical value for brain tumor classification. This supports the role of AI in aiding radiologists with faster and more reliable diagnoses. Looking forward, we intend to test our model on larger and more diverse datasets to further validate its reliability. We also plan to extend this research to other medical imaging domains, including X-rays, ultrasound, dermoscopy, endoscopy, and histopathology. Furthermore, we aim to incorporate transfer learning and fine-tuning with additional medical imaging datasets to explore improvements in generalization and scalability. The ultimate goal is to develop intelligent, end-to-end diagnostic tools that can be seamlessly integrated into clinical environments, contributing to improved patient care through early and accurate disease detection.

## ACKNOWLEDGEMENTS

I want to thank every author for helping collect, analyze, and interpret the data that allowed this study to be conducted.

### Funding

The authors received no funding for this work.

### Competing Interests

The authors declare that they have no competing interests.

### Author Contributions

- Adil H. Khan conceived and designed the experiments, performed the experiments, analyzed the data, performed the computation work, prepared figures and/or tables, authored or reviewed drafts of the article, and approved the final draft.
- Asad Khan conceived and designed the experiments, performed the experiments, analyzed the data, performed the computation work, prepared figures and/or tables, authored or reviewed drafts of the article, and approved the final draft.
- D.N.F Awang Iskandar conceived and designed the experiments, authored or reviewed drafts of the article, and approved the final draft.
- Hiren Mewada conceived and designed the experiments, performed the experiments, analyzed the data, authored or reviewed drafts of the article, and approved the final draft.
- Saqib Saeed conceived and designed the experiments, performed the computation work, authored or reviewed drafts of the article, and approved the final draft.
- Fahad Algarni conceived and designed the experiments, performed the computation work, authored or reviewed drafts of the article, and approved the final draft.
- Farhan Ullah conceived and designed the experiments, analyzed the data, authored or reviewed drafts of the article, and approved the final draft.

- Muhammad Asghar Khan conceived and designed the experiments, performed the experiments, prepared figures and/or tables, authored or reviewed drafts of the article, and approved the final draft.
- Naveed Iqbal conceived and designed the experiments, performed the experiments, prepared figures and/or tables, authored or reviewed drafts of the article, and approved the final draft.
- Ahmed A. Hussain conceived and designed the experiments, performed the experiments, analyzed the data, prepared figures and/or tables, authored or reviewed drafts of the article, and approved the final draft.

### Data Availability

The brain tumor dataset (from Nanfang Hospital and the General Hospital of Tianjin Medical University) is available at figshare: Cheng, Jun (2017). brain tumor dataset. figshare. Dataset. https://doi.org/10.6084/m9.figshare.1512427.v8.

The brain tumor MRI dataset is available at Kaggle: https://www.kaggle.com/datasets/masoudnickparvar/brain-tumor-mri-dataset.

The SARTAJ dataset is available at Kaggle: https://www.kaggle.com/sartajbhuvaji/brain-tumor-classification-mri.

The Br35H dataset is available at Kaggle: https://www.kaggle.com/datasets/ahmedhamada0/brain-tumor-detection?select=no.

### Supplemental Information

Supplemental information for this article can be found online at http://dx.doi.org/10.7717/peerj-cs.3154#supplemental-information.

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
