# Peer review of "BrainNet: a custom-designed CNN and transfer learning-based models for diagnosing brain tumors from MRI images"

_PeerJ Computer Science, doi:10.7717/peerj-cs.3154_

## Round 0.1 · original submission · Major Revisions

·

Basic reporting

The paper is well written, and deals with a classical brain tumor classification problems. It uses routine methods, and pre-trained models on three public datasets. The study is comprehensive but novelty is limited

Experimental design

Experiments are good, however comparision is not fair, comparision is made with methods trained on different datasets which is not correct.

Validity of the findings

The methods are validated on three datasets, with a minor increase in accuracy. Accuracy is more than 99.6%, which gives an impression of over-fitting.

Additional comments

The authors should try to (i) highlight the novelty (ii)make comparision fair and try more advanced models, such as Vision Transformers or others.

Reviewer 2 ·

Basic reporting

1. The manuscript is written in professional English, although some sentences, particularly in the abstract and introduction, could benefit from more concise phrasing.

2. The literature is comprehensive and well-referenced, though the literature review could be more concise by grouping related works (hybrid models, traditional ML approaches, CNN-based models).

3. Several figures and tables (diagrams of BrainNet-1/2/3, some result plots) are not referenced or explained in the body of the text.

Experimental design

1. The study does not evaluate the model’s performance on external or unseen datasets. Since two datasets were used the manuscript should include per-dataset evaluation to understand dataset-specific performance and enable fair comparison with prior works.
Please report the performance separately on FigShare and Kaggle datasets. Additionally, consider evaluating the model on an external dataset such as BraTS 2021 to assess robustness and generalizability.

2. It is unclear whether a model trained on one dataset can generalize to the other. This is critical to assess real-world applicability. Include an experiment where the model is trained on one dataset and tested on the other to evaluate domain transfer capability.

3. The training process was capped at 40 epochs, yet the training accuracy trend suggests further improvement was possible. Please explain why 40 epochs were chosen. Why not use 100 epochs with early stopping? Was the model still improving?

Validity of the findings

1. Evaluation is thorough using standard metrics with per-class results. The test accuracies are convincingly high, though BrainNet-3’s near perfect performance 99.92% raises concerns about overfitting or lack of external validation. Additional statistical significance testing, like confidence intervals, cross-dataset performance is recommended.

2. Table 8 compares BrainNet with other models that used different datasets and experimental protocols. This renders the comparison scientifically questionable. Revise Table 8 to compare with models that used the same dataset(s), ideally from the past three years, to ensure relevance and fairness.

Additional comments

This manuscript replicable and technically sound research study, even though it does not propose a novel algorithmic breakthrough.

The paper offers justified replication, expands on past works with practical dataset integration, and provides benchmarked evaluations that will be useful to researchers and practitioners working on medical AI applications.

With the suggested improvements, it can become a solid and reusable contribution to the brain tumor classification literature.

Reviewer 3 ·

Basic reporting

The authors work is not bad, but there’re are some issues that must be addressed. The following are the comments.
1. The key contribution of this work needs to be further highlighted as well as the technical flow should be refined.
2. Is the model inference fast enough for real-time or near-real-time usage? Can you please comment on the processing time of the model?
3. Is the framework compatible with deployment tools like ONNX, TensorRT, or TFLite for edge devices?
4. The literature review is weak. The authors should cite some more latest paper for comparison.
5. The quality of all the figures could be improved. Also, the text in the figures must be in the same style as the text in the paper body.
6. Are early stopping or learning rate scheduling used to prevent overfitting?
7. A table of parameters and a summary of the best model should be included.
8. "Figure 1 and Figure 2 appear stretched, distorting their clarity. Please resize them proportionally to ensure readability."
9. Replace all the words "TumorNet" with "BrainNet".
10. The existing references seem limited; citing the latest references is better for better comparison.
11. The English writing and grammar should be improved. Also, there are some typo errors. Please carefully revise the entire manuscript.

Experimental design

See above detailed comments

Validity of the findings

See above detailed comments

Additional comments

See above detailed comments

Reviewer 4 ·

Basic reporting

1. Some major editing is necessary: sections are written repetitively, model names aren’t constant (“BrainNet” is used instead of “TumorNet”) and several typos and formatting issues make reading tough.

2. Literature Context – Provides many references, but not enough comparison between them; significant work on MRI-tumor classification and attention-based CNNs is not thoroughly addressed.

3. Currently, the figures are not informative enough and parts of the paper about the datasets or models can be found in scattered or opposing places (for example, where the authors say they include three datasets, only two are discussed).

4. Source code, model weights and the precise divisions for training, validation and test were not supplied and so replicating the findings became difficult.

Experimental design

1. Problems with Understanding Dataset Information:

-> All of the following are not provided: sources, numbers of classes or how patients may overlap.
The explanations for bringing together FigShare and Kaggle images are not clear which could lead to unintentional leaks.

2. The Pipeline Provides Little Detail in the Pre-processing Step:
It is unnecessary to use both 224 × 224 and 256 × 256 resolutions.
Skull stripping and intensity normalization are not included in my procedure.

3. Database Splitting with Uncertainty and Data Expansion
Paper discusses using 70 / 10 / 20 training / validation / test set, even though later, it only mentions training and validation.
There is a risk that augmented versions of the same scan can be found in several train splits.

4. Single-Split Evaluation:
-> Practices only one random split, not k-fold cross-validation or repeat runs which lowers its rigor.
All validation was done using the same data used to train the model.

5. Drawing Baselines Is Often Unjust.
ResNet-34 was used with three classes, while BrainNet was tested with four, so hyper-parameter parity was not observed.
The process of adjusting and scheduling training activities for transfer-learning models is not widely reported.

6. No analysis of ablation or sensitivity.
No experiments were done that tested the importance of depth, dropout or attention modules in BrainNet-1/2/3.
Parameters, FLOPs and inference time are not available in the documentation.

7. The report has a lack of statistical rigor.
Data is not supported by confidence intervals, information on how runs differ or hypothesis testing (for example, McNemar).
Extremely high results have not been calculated relative to possible random variation.

Validity of the findings

1. The research is not well backed by statistics. Results showed very accurate findings with no confidence intervals, differences across runs or tests for statistical significance.

2. Comparing baselines: It’s not fair: ResNet-34 is built to recognize three things, but BrainNet is scored on four. Hyper-parameter tuning parity is not illustrated.

3. The situation where validation accuracy is lower than the training dataset and test accuracy is higher may mean there is leakage involved. Learning and ROC curves are not illustrated in the manuscript.

4. The authors say their network is state of the art and suitable for resource-shallow systems, but they do not back their statements up with technical details.

Additional comments

1. Dataset transparency
• Make a combined table including the sources, the number of patients in each class and the number of patients who appeared in both groups.
You should only accept augmented samples after merging into a split; make sure no patient occurs in more than one split.

2. Evaluation rigor
• Try both k-fold cross-validation and repeated random splits; show the average metric results with the standard deviation.
Statistics are useful for showing how your results are more beneficial than others.

3. Both architecture and ablation are essential parts of the process.
List all the details for each layer, indicate the number of parameters involved and present ablations on depth, dropouts and attention.
Tell me why you think the bespoke CNN is a better choice than some of today’s lightweight backbones.

4. Baseline fairness
All baseline methods should be trained on the same four-class task with the same datasets, augmentation and training schedules.
Make sure to note what hyper-parameters were used and how long it took to train each model.

5. Understanding how to read test results in light of clinical practice
• Offer arrangements of confusion matrices, ROC-AUC calculations and analyses of sensitivity and specificity.
• Talk about explainability (such as Grad-CAM), the risk of site biases and compatibility with the clinic.

6. Can results be easily repeated in the same setup?
• Upload your code, trained weights and reproduce scripts; state all random seeds.
• Make sure to include a statement about using ethical and confidential data.

---

## Round 0.2 · accepted · Accept

Thank you for your contribution.

Reviewer 3 ·

Basic reporting

No further comments.

Experimental design

-

Validity of the findings

-

Additional comments

-